# Impaired immune surveillance accelerates accumulation of senescent cells and aging

Yossi Ovadya[1], Tomer Landsberger[2], Hanna Leins[3,4], Ezra Vadai[1], Hilah Gal[1], Anat Biran[1], Reut Yosef[1], Adi Sagiv[1], Amit Agrawal[1], Alon Shapira[1], Joseph Windheim[1], Michael Tsoory[5], Reinhold Schirmbeck[4], Ido Amit [2], Hartmut Geiger[3,6] & Valery Krizhanovsky [1]

Cellular senescence is a stress response that imposes stable cell-cycle arrest in damaged cells, preventing their propagation in tissues. However, senescent cells accumulate in tissues in advanced age, where they might promote tissue degeneration and malignant transformation. The extent of immune-system involvement in regulating age-related accumulation of senescent cells, and its consequences, are unknown. Here we show that $Prf1^{-/-}$ mice with impaired cell cytotoxicity exhibit both higher senescent-cell tissue burden and chronic inflammation. They suffer from multiple age-related disorders and lower survival. Strikingly, pharmacological elimination of senescent-cells by ABT-737 partially alleviates accelerated aging phenotype in these mice. In $LMNA^{+/G609G}$ progeroid mice, impaired cell cytotoxicity further promotes senescent-cell accumulation and shortens lifespan. ABT-737 administration during the second half of life of these progeroid mice abrogates senescence signature and increases median survival. Our findings shed new light on mechanisms governing senescent-cell presence in aging, and could motivate new strategies for regenerative medicine.

[1] Department of Molecular Cell Biology, The Weizmann Institute of Science, 76100 Rehovot, Israel. [2] Department of Immunology, The Weizmann Institute of Science, 76100 Rehovot, Israel. [3] Institute of Molecular Medicine, Stem Cell and Aging, Ulm University, Ulm 89081, Germany. [4] Department of Internal Medicine I, University Hospital of Ulm, Ulm 89081, Germany. [5] Department of Veterinary Resources, The Weizmann Institute of Science, 76100 Rehovot, Israel. [6] Experimental Hematology and Cancer Biology Cincinnati Children's Hospital Medical Center, 45229 Cincinnati OH, USA. Correspondence and requests for materials should be addressed to V.K. (email: valery.krizhanovsky@weizmann.ac.il)

Aging is characterized by a functional decline in many physiological systems. Both environmental and endogenous stressors, might drive the progression of physiological aging[1–3]. These include telomere attrition, genomic instability, epigenetic alterations, and loss of proteostasis, each of which damages cells and compromises their functionality[3]. Cellular senescence, a central component of aging, is a cell-intrinsic stress response programmed to impose stable cell-cycle arrest in damaged cells, thus preventing them from propagating further damage in tissues[2,4,5]. Normally, a sequence of events leads to the clearance of senescent cells and allows regeneration of the tissues that harbor them[6–9]. In advanced age, however, the efficiency of this process may be compromised, as suggested by the tendency of senescent cells in the tissues of old individuals to accumulate[10,11]. This accumulation is reportedly conserved across different species, including rodents[12,13], primates[14,15], and humans[10,11]. Under such conditions, the beneficial cell-autonomous role of senescence might be outstripped by a negative impact of senescent cells on other cells, an effect mediated via the senescence-associated secretory phenotype (SASP), which has marked pro-inflammatory characteristics[16].

Senescent cells are subject to immune surveillance by multiple components of the immune system[6–9,17]. Senescent cells attract and activate immune cells and serve as highly immunogenic targets for immune clearance. The immune response against senescent cells varies between different pathological conditions. For example, in fibrotic liver senescent cells derived from activated hepatic stellate cells are cleared by natural killer (NK) cells[7], whereas senescent pre-malignant hepatocytes are eliminated via the adaptive immune system[9]. In other pathological conditions, for example in the case of dysplastic nevi, immune clearance does not occur and senescent cells persist for years[18]. In the context of aging, it is not known to what extent the immune system participates in regulating the number of senescent cells, and whether age-related impairment of immune function contributes to the accumulation of senescent cells in old individuals[19].

Perforin, a pore-forming protein found in intracellular granules of effector immune cells, is an important mediator of immune cytotoxicity[20,21]. Upon degranulation, perforin-formed pores enable granzyme penetration and caspase activation to induce apoptosis of the target cell. Perforin-mediated granule exocytosis (but not death-receptor-mediated apoptosis) is essential for the immune surveillance of senescent cells, and disruption of this pathway leads to the accumulation of senescent cells in damaged livers[22]. To investigate the consequences of impaired immune surveillance of senescent cells in aging, we followed the aging process in mice in which granule-exocytosis-mediated apoptosis was disabled as a result of perforin gene knockout ($Prf1^{-/-}$)[23].

Our data indicates that compared to wild-type (WT) mice, $Prf1^{-/-}$ mice accumulates more senescent cells in their tissues with age. The accumulation of senescent cells in these $Prf1^{-/-}$ mice is accompanied by a progressive state of chronic inflammation, followed by increased tissue fibrosis and other types of tissue damage, as well as compromised organ functionality. The poor health of old $Prf1^{-/-}$ mice is associated with fitness reduction, weight loss, kyphosis, older appearance, and shorter lifespan than that of WT controls. These deviant phenotypes were also observed in $Prf1^{-/-}$ mice crossed with progeroid mice. Elimination of senescent cells from old $Prf1^{-/-}$ mice can be achieved by pharmacological inhibitors of the BCL-2 family of proteins, such as ABT-737. This pharmacological approach attenuates age-related phenotypes and gene expression profile in $Prf1^{-/-}$ mice. Finally, implementation of this approach on $Prf1^{-/-}$ progeroid mice increases median lifespan of these mice.

## Results

**Perforin deficiency accelerates senescence with age.** The prevalence of senescent cells in tissues increases with chronological age[10,11]. While senescent cells are subjected to immune cell cytotoxicity, it is not clear whether age-related impaired cell cytotoxicity could account for their accumulation. To examine this possibility, we set an in vivo experiment for assessment of systemic cytotoxicity of CD8$^+$ T cells in young and old mice. The systemic cytotoxicity of CD8$^+$ T cells in vivo was reduced more then 3-fold ($P < 0.01$) in aged mice compared to the young ones (Supplementary Figure 1). Age-related decline in cell cytotoxicity was shown to be a consequence of reduced release and binding of perforin at the immunological synapse[24]. To determine whether immune cell cytotoxicity plays a role in regulation of tissue burden of senescent cells throughout aging, we used $Prf1^{-/-}$ mice, in which immune surveillance of senescent cells is impaired[22]. We established cohorts of $Prf1^{-/-}$ and control WT mice, both on the background of C57BL/6, and examined selected organs including livers, pancreas, lungs, and skin in 2, 12, and 24-month old mice (defined hereafter as "young", "adult", and "old", respectively). To assess time-dependent accumulation of senescent cells in those tissues, we first assayed them for senescence-associated-β-galactosidase (SA-β-Gal) activity, an assay commonly used to identify senescent cells in tissues and in culture[10]. We observed an increase in the number of SA-β-Gal + cells with age in all tissues examined. Increase was more pronounced in the $Prf1^{-/-}$ mice (Fig. 1a, b, Supplementary Figure 2a). Quantitative analysis of these cells in WT mice indicated that they comprise around 10% of the examined tissues by the time these mice reach 24 months of age. At the same age in $Prf1^{-/-}$ mice those cells comprised up to 43% of the total cells, demonstrating a significant ($P < 0.005$) increase of 2- to 4-fold (depending on the tissue) compared to WT mice (Fig. 1b). These finding indicate that tissues of old $Prf1^{-/-}$ mice extensively accumulate SA-β-Gal + cells.

The most established molecular marker of senescence and a key component of the senescence program is p16 ($Cdkn2a$)[25]. Accumulation of senescent p16-positive cells shortens mouse lifespan (Baker et al, 2016). We studied the correlation of SA-β-Gal activity in the liver with the pattern of p16 expression. The dynamics of p16 expression mimicked the levels of SA-β-Gal + cells (Fig. 1c–e). The expression of p16 assessed by immunohistochemistry (Fig. 1c, d) or by quantitative RT-PCR (Fig. 1e) increased with age in WT mice, while $Prf1^{-/-}$ old mice had a significant increase in p16 expression compared to WT of the same age. Moreover, expression of p16 overlapped substantially with SA-β-Gal activity in the livers of old $Prf1^{-/-}$ mice, mostly in non-hepatocytes cells (Fig. 1f). Therefore, both p16-positive and SA-β-Gal-positive cells accumulate more extensively in the liver of $Prf1^{-/-}$ mice compared to WT mice.

To achieve a more reliable quantification of senescent cells in tissues, we applied a method based on a combination of SA-β-Gal and molecular markers of senescence on a single-cell level[26]. One such marker is loss of the nuclear high-mobility group box 1 protein (HMGB1)[27]. We therefore studied the prevalence of SA-β-Gal + /CD45$^-$/HMGB1$^-$ cells as a cell population representative of tissue-resident senescent cells by the quantitative single-cell based method and visualized by the ImageStreamX apparatus which combines flow cytometry and microscopy (Fig. 1g). After validating the presence of the SA-β-Gal + populations in the liver, pancreas, and lung (Supplementary Figure 2b), we analyzed the nuclear HMGB1 staining in CD45$^-$/SA-β-Gal + cells. While nuclear HMGB1 is ubiquitously expressed in the tissues examined, most of CD45$^-$/SA-β-Gal + cells were found to be negative for nuclear HMGB1 staining (Fig. 1h, Supplementary Figure 2c). The presence of the SA-β-Gal + /CD45$^-$/HMGB1$^-$

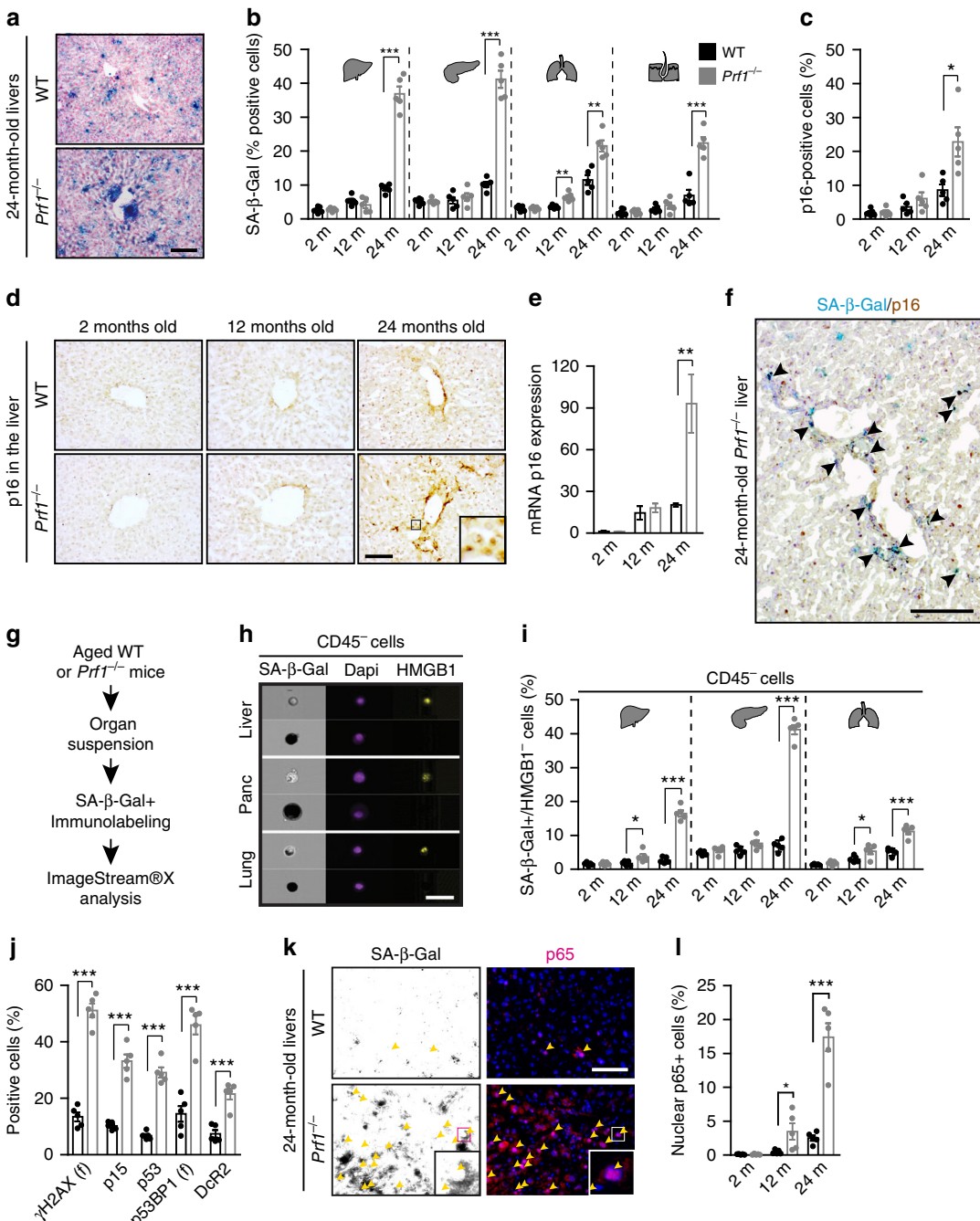

**Fig. 1** Old *Prf1*⁻/⁻ mice accumulate more senescent cells then old WT mice. Cohorts of *Prf1*⁻/⁻ and wild type (WT) C57BL/6 female mice at the age of 2, 12, and 24 months were sacrificed and their livers, pancreas, lungs, and skin were examined for the presence of senescent cells. **a** SA-β-Gal activity representative frozen sections of livers from 24-months-old mice. Scale bar, 100 μm. **b** Quantification of cells with marked SA-β-Gal activity, based on Nuclear Fast Red counterstaining, in liver, pancreas, bronchial epithelia, and skin epidermis. ($n = 5$ mice per group). **c** Percentages of cells stained positively for p16 in the liver. Values are means ± SEM, ($n = 5$ mice per group). **d** Representative images of IHC staining for p16 on liver frozen section. Scale bar, 100 μm. **e** Real-time qPCR analysis for expression of p16 in livers. Values are means ± SEM, ($n = 5$ mice per group). **f** Representative image of SA-β-Gal activity combined with IHC staining for p16 on liver frozen section from 24 old *Prf1*⁻/⁻ mice. Scale bar, 100 μm. **g** Scheme of examination of senescence by ImageStreamX. **h** Representative ImageStreamX images of CD45⁻ negative cells derived from indicated organs that were stained for HMGB1 and SA-β-Gal. Scale bar, 50 μm. **i** Quantification of CD45⁻/SA-β-Gal + /HMGB1⁻ population in each organ, as analyzed in (**h**). ($n = 5$ mice per group). **j** Percentages of cells stained positively for γH2AX, p15, p53, p53BP1, and DcR2 in livers from old mice. ($n = 5$ mice per group). **k** Immunofluorescence staining for p65 (pink) in SA-β-Gal + cells (indicated by arrows in black/white photos) in livers from old mice. Scale bar, 50 μm. **l** Percentages of nuclei positive for p65 in livers. For all graphs, values are means ± SEM, ($n = 5$ mice per group). Student's *t*-test was used for all comparisons between Prf1⁻/⁻ and WT female mice (*$P < 0.05$, **$P < 0.01$, ***$P < 0.001$)

cells was increased in an age-dependent manner in both groups with a significant ($P < 0.05$), 2- to 6-fold increase in old $Prf1^{-/-}$ mice compared to old WT mice (Fig. 1i). We also examined the expression of an additional set of senescence markers, previously used to identify senescent cells[28], comprised of $\gamma$H2AX foci, p15, p53, p53BP1 foci, and DcR2, in the tissues. A marked increase in expression all of those proteins was observed in old $Prf1^{-/-}$ mice compared to the old WT mice and it was overlapping with SA-$\beta$-Gal staining of the consecutive sections (Supplementary Figure 2d). Quantitative analysis of each of these markers showed a significant ($P < 0.0001$) increase in positive cells in the old $Prf1^{-/-}$ mice (Fig. 1j). Apparently, expression of a variety of senescence markers increases in old $Prf1^{-/-}$ mice.

Due to their pro-inflammatory nature, accumulation of senescent cells in tissues can potentially drive age-related chronic inflammation[29–31]. NF-$\kappa$B pathway is one of the main regulators of the pro-inflammatory profile of senescent cells[2,32–34]. The p65 subunit of NF-$\kappa$B (also known as a RelA) translocates to the nucleus to drive transcriptional regulation. Co-staining of liver sections with p65 and SA-$\beta$-Gal demonstrated that SA-$\beta$-Gal + cells in the livers of old $Prf1^{-/-}$ mice display nuclear p65 frequencies that are 10-fold higher than in the livers of old WT mice, reflecting activation of the NF-$\kappa$B pathway in these cells in $Prf1^{-/-}$ mice (Fig. 1k). Quantification of nuclear p65 + cells in liver sections showed age-dependent increase, which was further elevated in old $Prf1^{-/-}$ mice compared to the old WT mice (Fig. 1l). Taken together, our findings indicate that $Prf1^{-/-}$ leads to increased accumulation of senescent cells in tissues with age, as identified by multiple markers representing different characteristics of the senescence phenotype.

**Perforin deficiency drives age-dependent chronic inflammation.** Chronic inflammation is one of the characteristics of aging that are associated with the presence of senescent cells[3,16,34]. We evaluated the expression of known SASP components in the liver tissue of young, adult, and old WT and $Prf1^{-/-}$ mice by qPCR analysis (Fig. 2a). The expression of Il-6, RANTES, and JE was significantly increased in old $Prf1^{-/-}$ mice compared to old WT mice, while other cytokines show a similar tendency. Together with the increase in the cytokine expression, the number of tissue-infiltrating immune cells gradually increased with a similar kinetic to accumulation of senescent cells (Fig. 2b). This immune infiltration was further escalated in old $Prf1^{-/-}$ mice compared to WT ($P < 0.05$), in a manner that was largely represented by infiltration of T and NK cell subsets (Fig. 2b). Histological analysis verified the existence of accumulating immune cells in different sites of $Prf1^{-/-}$ mice (Supplementary Figure 3a). Examination of the liver cryosections from old $Prf1^{-/-}$ mice revealed multiple white dots that were barely noticeable in the livers from old WT mice (Fig. 2c, left panel). These structures appeared to be occupied by CD45$^+$ cells and were therefore suspected to be immune foci. On examining these foci for the presence of different subsets of immune cells, we found that they were dominated by CD3$^+$ cells and NK1.1$^+$ cells (Fig. 2c, right and Fig. 2d). The robust infiltration observed in livers of the $Prf1^{-/-}$ mice was accompanied by an increase of 1.5 fold in their liver weight compared to the WT (Supplementary Figure 3b, c).

To determine whether the increased accumulation of senescent cells and immune cells in tissues of $Prf1^{-/-}$ mice are accompanied by systemic inflammation, we used a cytokine array to compare their serum cytokine levels with those of WT mice at the age of 2, 12, and 24 months. Interestingly, at the age of 2 months there was no difference between $Prf1^{-/-}$ and WT mice, and at the age of 12 months only slight elevations were detected in the $Prf1^{-/-}$ mice. Nevertheless, in 24-months-old $Prf1^{-/-}$

mice we found a strong upregulation of pro-inflammatory factors (such as RANTES, TNF-$\alpha$, IP-10, and MIG), in addition to an upregulation of some anti-inflammatory factors (such as IL-10 and IL-1RA) (Fig. 2e, Supplementary Figure 3d). Furthermore, the white blood counts (WBCs) of the old $Prf1^{-/-}$ mice were more then 2-fold higher than those of the old WT mice ($P = 0.014$, Fig. 2f). In addition to a raised WBC, chronic inflammation usually leads to an increase in spleen size. The spleens of old $Prf1^{-/-}$ mice were larger, both in size and in weight, than in their WT counterparts (Fig. 2g, h, Supplementary Figure 3e). In order to better characterize the inflammatory situation in the $Prf1^{-/-}$ mice throughout aging, we looked for the presence of inflammatory cells in some of their internal organs. Flow-cytometry analysis of liver, pancreas, and lung of young $Prf1^{-/-}$ mice did not show differences in their immune composition at young age comparing to the WT (Supplementary Figure 3f). Our observations thus suggest that accumulation of senescent cells in tissues of $Prf1^{-/-}$ mice is accompanied by an increased inflammatory response, which might facilitate the establishment of age-related chronic inflammation.

**Perforin deficiency promotes age-related disorders and death.** The establishment of chronic inflammation, also known as "inflammaging", is a major risk factor for both morbidity and mortality in elderly people[16,35]. Owing to the increase in general inflammation in old $Prf1^{-/-}$ mice, we set to monitor physiological integrity of organs in these mice compared to WT mice. We examined different organs by pathological analysis of hematoxylin and eosin (H&E)-stained sections. Our analysis demonstrated an acceleration of age-dependent general impairment of tissues in the $Prf1^{-/-}$ mice compared to the WT mice (Supplementary Figure 4a). One manifestation of tissue impairment is the abnormal accumulation of extracellular matrix in tissues, commonly called fibrosis. In old $Prf1^{-/-}$ mice, fibrotic areas in the liver, pancreas, skin, and kidney were extensively formed, and were increased more than 2-fold compared to old WT controls ($P < 0.01$) (Fig. 3a, Supplementary Figure 4a). Additionally, in the kidney of $Prf1^{-/-}$ mice, periodic acid–Schiff (PAS) staining revealed enhanced formation of glomerular sclerosis compared to WT controls ($P < 0.01$, Fig. 3b, c, Supplementary Figure 4b). In the skin of old $Prf1^{-/-}$ mice, loss of the subcutaneous fat layer and a moderate attrition of hair follicles have resulted in a significantly reduced skin thickness, compared to the old WT mice ($P = 0.005$, Fig. 3d, e). Skin abnormalities in old $Prf1^{-/-}$ mice also resulted in significant reduction in hair density ($P = 0.004$), and a 20-fold increase in the prevalence of gray hair (Fig. 3f, g). Due to these phenotypes, old $Prf1^{-/-}$ mice appeared visibly older than their WT counterparts (Fig. 3h). To investigate whether the described tissue impairments are affecting the physiological condition of old $Prf1^{-/-}$ mice, we examined their blood for markers widely used to assess functions of different organs. Blood tests indicated a significant increase in general markers of tissue damage in internal organs, including the liver (ALP, AST, and ALT), pancreas (amylase), lungs (LDH), and muscles (CPK) (Fig. 3i). Analysis of markers in the blood also showed that urea levels in the blood of old $Prf1^{-/-}$ mice were significantly higher than in old WT ($P = 0.016$; Fig. 3i). This might be a consequence of a reduction in kidney's filtration capacity and glomerular sclerosis (Fig. 3b, c), frequently observed in both humans and mice of advanced age[36,37]. Apparently, therefore, accumulation of senescent cells in perforin-deficient mice is associated with impaired tissue structure and function of multiple organs.

The progressive decline in cellular function that occurs during aging ultimately affects the fitness on the organism level[38]. For

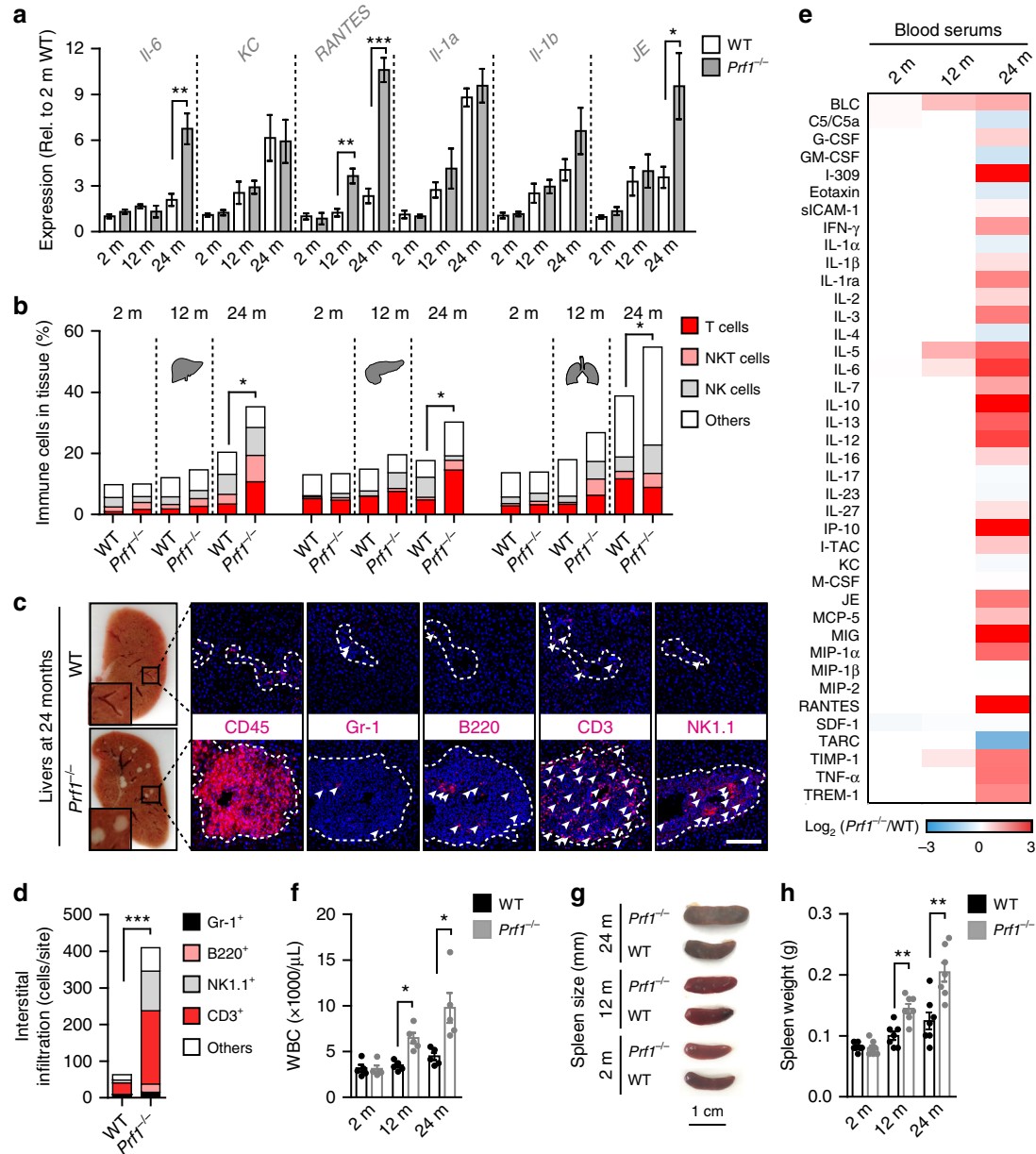

**Fig. 2** $Prf1^{-/-}$ mice develop chronic systemic and local inflammation. **a** Real-time qPCR analysis for expression of p16 and SASP components in livers from 2, 12, and 24-months-old $Prf1^{-/-}$ and WT female mice. ($n = 5$ mice per group). **b** Flow-cytometric quantification of total immune cells (CD45$^+$), T cells (CD45$^+$/CD3$^+$/NK1.1$^-$), NKT cells (CD45$^+$/CD3$^+$/NK1.1$^+$), and NK cells (CD45$^+$/CD3$^-$/NK1.1$^+$) in livers, pancreas, and lungs of 2, 12, and 24-months-old $Prf1^{-/-}$ and WT mice. ($n \geq 5$ mice per group). **c** Immune foci observed (indicated in squares, left panel) in liver cryosections from old $Prf1^{-/-}$ and WT mice. The immune foci were analyzed on sections stained immunofluorescently for CD45, Gr-1, B220, CD3, and NK1.1. Scale bar, 100 μm. **d** Numbers of the different immune subsets in livers from old $Prf1^{-/-}$ mice and WT mice, based on the immunofluorescence staining presented in (**c**). **e** Array of serum cytokine levels in 2, 12, and 24-months-old $Prf1^{-/-}$ female mice relative to WT female mice at the corresponding age. Colors represent increase (red) or decrease (blue) in the average intensity of 3 pooled samples. Values are shown in log2 according to the legend panel. **f** White blood cell counts of 2, 12, and 24-months-old $Prf1^{-/-}$ and WT female mice. ($n = 5$ mice per group). **g** Representative photos of spleens from 2, 12, and 24-months-old $Prf1^{-/-}$ and WT female mice. **h** Spleen weights of 2, 12, and 24-months-old $Prf1^{-/-}$ and WT female mice. ($n = 7$ mice per group). For all graphs, values are means ± SEM. Student's t-test was used for all comparisons between $Prf1^{-/-}$ and WT mice. (*$P < 0.05$, **$P < 0.01$, ***$P < 0.001$)

example, old mice, like old humans, tend to be less active than young ones. To evaluate the overall fitness of $Prf1^{-/-}$ mice we tested their voluntary exercise, as well as their muscle strength and coordination. We found that old $Prf1^{-/-}$ mice showed a reduction of more than 2-fold in voluntary exercise ($P < 0.05$; Fig. 3j, Supplementary Figure 5a), and their grip strength and coordination in a "hang-wire" test were significantly decreased, compared to old WT mice ($P = 0.044$; Fig. 3k). Mammals also tend to lose weight during late stages of life[39]. Old $Prf1^{-/-}$ mice

also suffered from progressive weight loss (Fig. 3l), which was comprised of both muscle loss and fat loss (Supplementary Figure 5b). Another aspect of organism-level fitness reduction is kyphosis, an exaggerated rounding of the spine. Old $Prf1^{-/-}$ mice exhibited a higher incidence of age-related kyphosis than old WT mice ($P < 0.05$; Fig. 3m). An age-related pathology reported to be common in C57BL/6 males is enlargement of the seminal glands[39,40]. Interestingly, many of the old $Prf1^{-/-}$ males in our study had enlarged seminal glands (in 70% of $Prf1^{-/-}$ mice

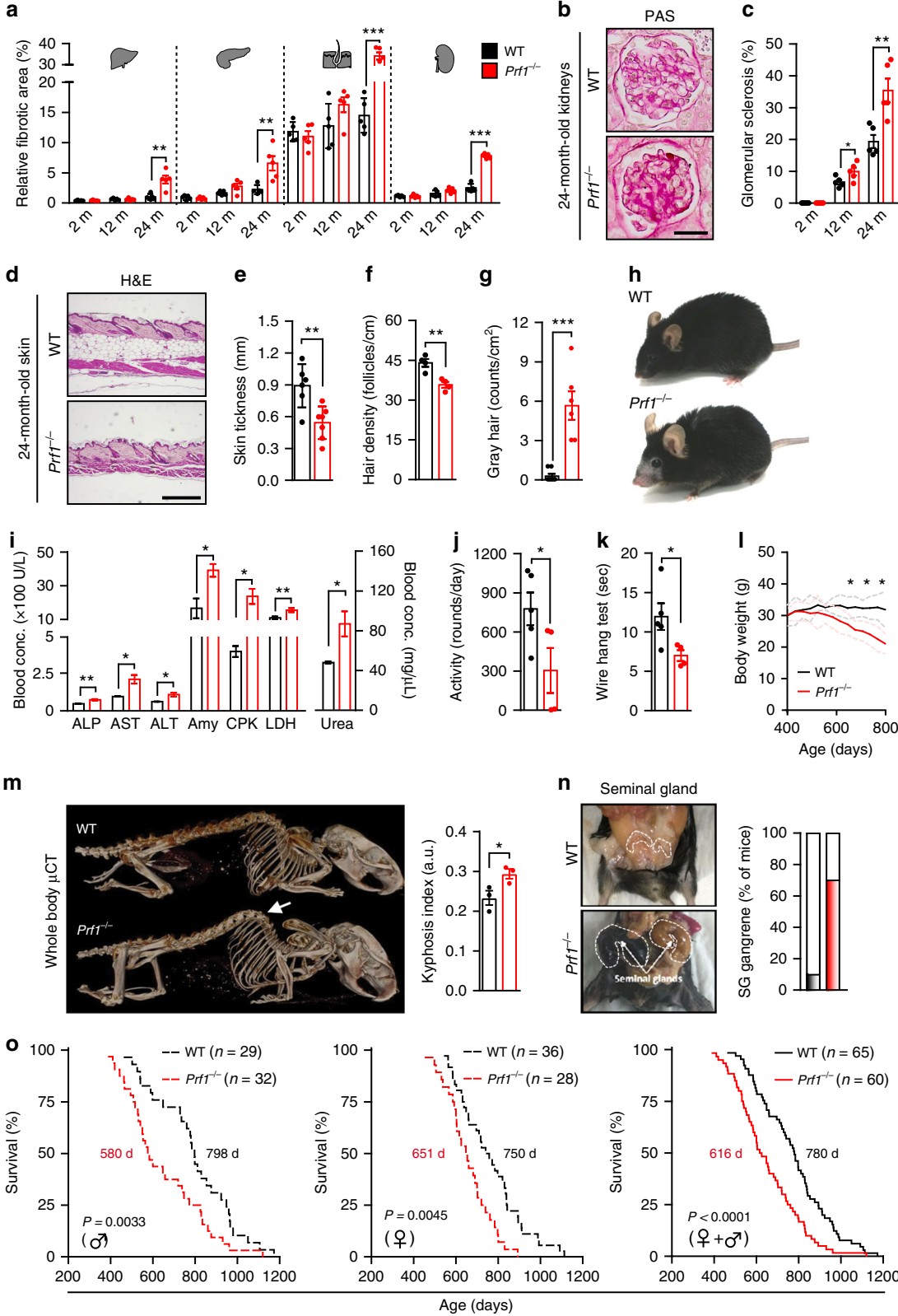

compared to 10% in WT), which eventually resulted in gangrene (Fig. 3n). These glands were enriched with SA-β-Gal + cells in the *Prf1*⁻/⁻ mice, and the thickness of their fibrotic layer was 3-fold higher in these mice than in the old WT (Supplementary Figure 5c, d). Therefore, our finding showed *Prf1*⁻/⁻ mice develop multiple physiological disorders usually associated with older age, which could lead to shorten lifespan. We monitored the

natural lifespan of cohorts of *Prf1*⁻/⁻ and WT mice from both genders. The median survival of *Prf1*⁻/⁻ mice was shorter (616 days) than that of WT mice (780 days), owing to the higher rate of early deaths, starting from the age of 14 months (Fig. 3o). Notably, the decreased median lifespan of *Prf1*⁻/⁻ mice was evident in both males and females (Fig. 3o). Overall, perforin deficiency is associated with accumulation of senescent cells,

**Fig. 3** $Prf1^{-/-}$ mice exhibit reduced fitness, higher rates of age-related disorders and a shorter lifespan. Cohorts of $Prf1^{-/-}$ and WT male mice were followed for survival and analyzed at the age of 2, 12, and 24 months. **a** Percentages of fibrotic areas in indicated tissues based on the Sirius red-staining in Supplementary Figure 4a. ($n \geq 5$ mice per group). **b** PAS staining of glomeruli from kidneys of 24-months-old $Prf1^{-/-}$ and WT mice. Scale bar, 25 μm. **c** Percentages of sclerotic glomeruli, based on PAS-stained kidney sections. ($n = 5$ mice per group). **d** H&E-stained sections of lower-back skin. Scale bar, 200 μm. **e** Quantification of skin thickness of old mice. ($n \geq 6$ mice per group). **f** Hair density on the lower back of old male mice. ($n = 4$ mice per group). **g** Numbers of gray-hair follicles per square centimeter of skin on the lower back of old mice. ($n \geq 6$ mice per group). **h** Representative photos of the old mice with considerable differences in skin and in hair between the two genotypes. **i** Levels of common markers of damage in the sera of old mice. ($n = 3$ mice per group). **j** Average activity levels based on voluntary exercise of old mice. ($n \geq 4$ mice per group). **k** Grip strength analysis based on a hang-wire test in old mice. Values indicate time that a mouse managed to hold on the wire. ($n \geq 4$ mice per group). **l** Weight curves of $Prf1^{-/-}$ and WT male mice. Values are means, dashed lines represent ± SEM ($n \geq 20$ mice per group). **m** μCT-3D images of old mice. The arrow indicates dorsal kyphosis. Kyphotic index, ($n = 3$ mice per group). **n** The prevalence of seminal gland (shown) gangrene in old $Prf1^{-/-}$ and WT male mice. **o** Kaplan–Meier curves for $Prf1^{-/-}$ and WT mice. Chi square Gehan–Breslow-Wilcoxon test was used for statistical analysis. For all graphs, values are means ± SEM. Student's $t$-test was used for all other comparisons between $Prf1^{-/-}$ mice and WT mice. (*$P < 0.05$, **$P < 0.01$, ***$P < 0.001$)

increase in total body inflammation, impaired function of multiple organs, and reduced survival.

**ABT-737 alleviates age-related phenotype of $Prf1^{-/-}$ mice.** Clearance of p16$^{Ink4a}$-positive senescent cells in mice extends their health-span[36,41]. To examine whether clearance of senescent cells could counteract age-related phenotype caused by impaired immune cell cytotoxicity, we administered the previously reported senolytic drug ABT-737[42]. Senescent cell viability is dependent on the expression of anti-apoptotic proteins from the BCL-2 family[42–46]. Accordingly, treatment with their specific inhibitors, such as ABT-737 or ABT-263, skews senescent cells toward apoptosis both in vitro and in vivo[42–44]. We administrated ABT-737 (25 mg/kg) or a vehicle solution to 18-months-old $Prf1^{-/-}$ mice, for two consecutive days monthly for a period of two months (Fig. 4a). Two weeks after the first ABT-737 injection, voluntary activity of ABT-737 treated mice increased compared to control mice (Fig. 4b). While the activity of the control group decreased over the two months as expected from aged mice, the ABT-737-treated group increased their activity 2-fold at the end of the first month and this increase was retained till the end of the experiment ($P = 0.026$; Fig. 4c). At the end of the two-month period, the amount of senescent cells was reduced in the ABT-737 treated mice as evaluated by SA-β-Gal staining (Fig. 4d) and by quantitative analysis of SA-β-Gal + /CD45-/HMGB1$^-$ cells in different tissues (Fig. 4e). To examine whether reduction on senescent cell burden was accompanied by a reduction of the hyper-inflammatory profile of $Prf1^{-/-}$ mice, we tested their serum by cytokine array. Interestingly, the top five proteins that were reduced by ABT-737 administration (Fig. 4f, Supplementary Figure 6a) include some of the most elevated proteins in the serum of old $Prf1^{-/-}$ mice compared to WT mice (Fig. 2e). Moreover, both WBCs counts and spleen weight were significantly reduced in ABT-737 treated group ($P = 0.017$ and $P = 0.039$, respectively, Fig. 4g, h). In line with this effect, ABT-737 treatment led to reduced immune infiltration into tissues (Fig. 4i), and reduced fibrotic area (Fig. 4j, Supplementary Figure 6b). In order to evaluate the effect of ABT-737 on the tissues by an independent approach, we systematically characterized the mRNA expression profile of ABT-737-treated old $Prf1^{-/-}$ mice and vehicle-treated old $Prf1^{-/-}$ mice using genome wide RNA sequencing (RNA-seq). Differential expression analysis and subsequent functional (Gene Ontology) analysis on kidney, liver, lung, and skin indicates a general repression of immune response, cytokine production, and endocytosis ($P = 1.4e^{-92}$; $1.2e^{-25}$ and $2.3e^{-13}$, respectively; Fig. 4k, Supplementary Data 1). The analysis also revealed several upregulated processes including fatty acid metabolism, cell morphogenesis, and development ($P = 6.6e^{-8}$; $3e^{-5}$ and $3.1e^{-4}$, respectively; Fig. 4k, Supplementary Data 1). Encouraged by significant downregulation in cytokine

production, we evaluated ABT-737 effects on a more confined SASP signature[47] in these tissues. Out of 56 established SASP factors, we detected 40 factors in at least one of the tissues, while different tissues express from 20 to 34 of the factors (Fig. 4l). In all four tissues examined, affected genes were enriched for SASP factors (hypergeometric test $P < 0.001$). To assess whether ABT-737 treatment drives gene expression levels towards those of young mice, we performed RNA-seq on the same tissues from 3-month-old $Prf1^{-/-}$ mice. We found that expression levels of genes affected by ABT-737 were driven towards the levels in young mice (Fig. 4l, Supplementary Figure 6c). ABT-737 treatment positioned overall SASP expression levels of old $Prf1^{-/-}$ mice closer to those of young $Prf1^{-/-}$ mice in kidney, liver, and skin ($P < 0.01$, $P < 0.05$, $P < 0.001$, respectively; Fig. 4l).

Effects observed in SASP factors point to the possibility of global shift of the transcriptional profile to a younger state following the ABT-737 treatment. Indeed, 74% of genes affected by ABT-737 treatment were also affected by aging (Fig. 4m, upper panel). This significantly larger-than-random (Hypergeometric test, $P < 2e^{-16}$) overlap suggests an interaction between ABT-737 treatment and age-related transcriptional changes. Importantly, ABT-737 effect counteracts the effect of aging on the vast majority of these genes, literally rejuvenating the expression profile of the mice. This effect becomes clear when the data are presented by regression of gene expression log fold-change old ABT-737/Young on old Vehicle/Young (Fig. 4m, lower panel, Supplementary Data 2) for all age-affected genes. The linear regression line (dashed line, Fig. 4m, lower panel, Supplementary Data 2) has a tilt < 1, indicated that the expression profile of old ABT-737 treated mice is more similar to young compared to old vehicle-treated mice ($P < 5e^{-5}$). This result is further supported by correlation analysis, showing that log(old Vehicle/Young) anti-correlates with log(ABT-737/Vehicle) (Spearman correlation test $\rho = -0.63$, $-0.46$, $-0.59$, and $-0.25$ for kidney, liver, lung, and skin respectively, $P < 2e^{-16}$). Overall, our results show that the senolytic drug ABT-737 eliminates senescent cells and counteracts age-related phenotype caused by $Prf1$ gene knockout in old mice.

**ABT-737 extends the lifespan of progeroid mice.** In some cases of accelerated aging, senescent cell formation can be enhanced by a repetitive intrinsic stress; that appears to occur in Hutchinson–Gilford progeria syndrome (HGPS), where accumulation of the progerin protein causes defects of the nuclear lamina and frequent damage to DNA[48,49]. $LMNA^{+/G609G}$ mouse harbors a point mutation in the $LMNA$ gene, which is identical to the HGPS causing mutation[50]. This HGPS mouse model demonstrates an extensive accumulation of senescent cells (Supplementary Figure 7a, b)[50]. To inquire the possibility that pharmacological elimination of senescent cells could also increase the

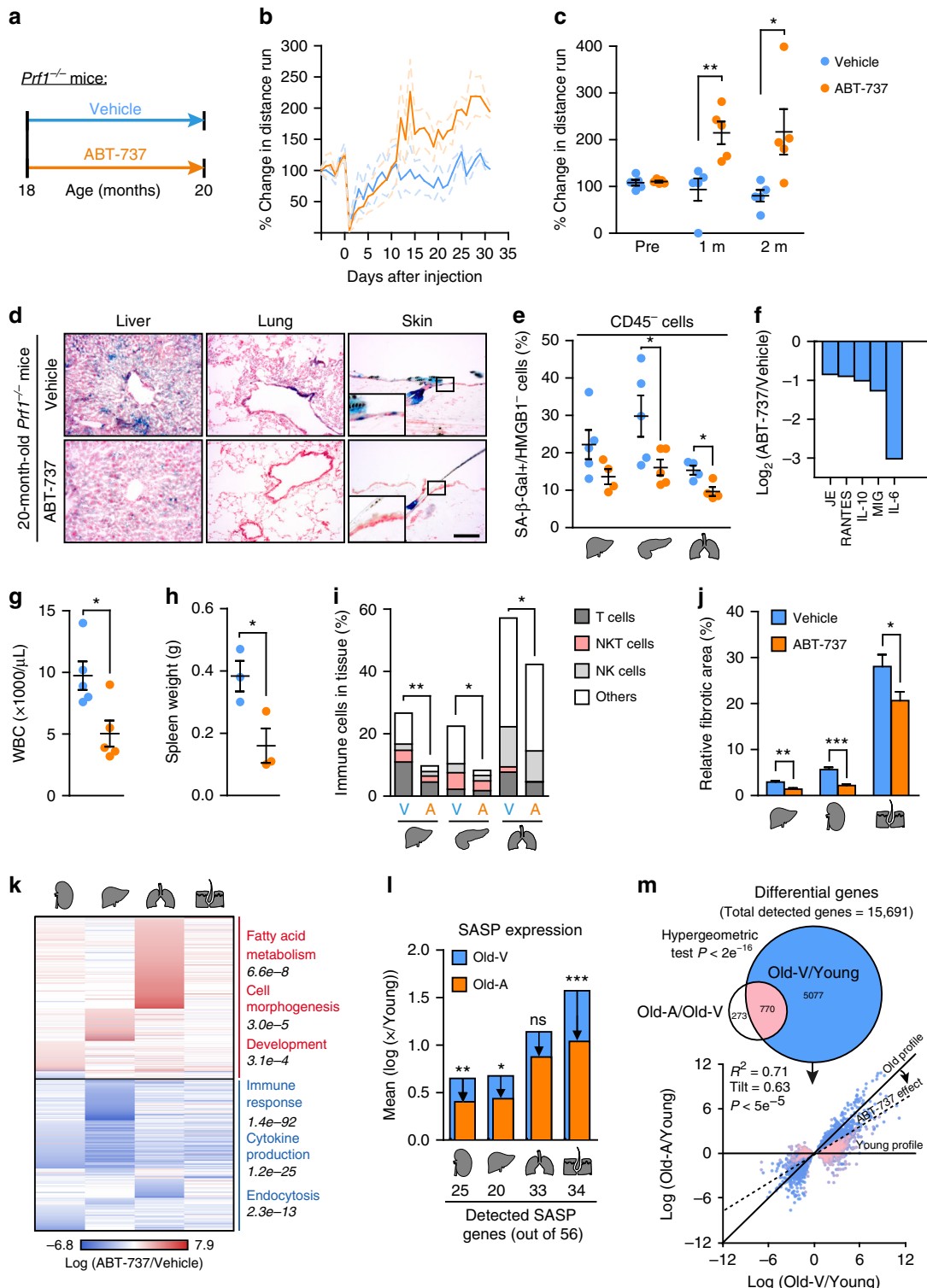

lifespan of progeroid mice, we have tested the effect of the senolytic drug ABT-737 on *LMNA*[+/G609G] mice. *LMNA*[+/G609G] mice were treated with ABT-737 for two consecutive days monthly from the age of 7 months and their tissues were analyzed at 11 months (Fig. 5a). We quantified the SA-β-Gal + cells present in the livers, pancreas, lungs, and skin of *LMNA*[+/G609G] mice. ABT-737 treatment resulted in a substantial decrease in SA-β-Gal + cells in all of the tested tissues (Fig. 5b), with an efficacy of elimination ranging from 50% (liver) to 70% (lungs) (Fig. 5c). Elimination of senescent cells was further validated by an overall

reduction in molecular markers of senescence (γH2AX, p15, p53, and p53BP1) (Fig. 5d, Supplementary Figure 7c). Therefore, treatment with ABT-737 resulted in considerable decrease in the numbers of senescent cells in tissues of *LMNA*[+/G609G] mice.

The accelerated aging phenotype in *LMNA*[+/G609G] mice has been previously linked to an NF-κB-mediated systemic inflammatory response[50], similarly observed in *Prf1*[−/−] mice. To determine whether the reduced tissue burden of senescent cells following ABT-737 treatment is associated with reduced inflammatory profile in old *LMNA*[+/G609G] mice, we evaluated

**Fig. 4** Treatment with ABT-737 counteract accelerated aging process of $Prf1^{-/-}$ male mice. **a** Starting from 18 months, ABT-737 or vehicle were administered to $Prf1^{-/-}$ mice at the beginning of each month, the mice were analyzed at the age of 20 months. **b** Change in distance run on a voluntary exercise test. Values are means and dashed lines represent ± SEM for each curve ($n = 5$ mice per group). **c** Average voluntary exercise levels during the last 3 days of the two months ($n = 5$ mice per group). **d** SA-β-Gal activity in indicated organs. Scale bar, 100 μm. ($n = 5$ mice per group). **e** CD45⁻/SA-β-Gal + /HMGB1⁻ population in indicated organs ($n \geq 4$ mice per group). **f** Relative cytokine levels in the serum for top five altered cytokines. Bars represent average intensity of 3 pooled samples. **g** WBC counts. ($n = 5$ mice per group). **h** Spleen weights. ($n = 3$ mice per group). **i** Flow-cytometry quantification of total immune cells (CD45⁺), T cells (CD45⁺/CD3⁺/NK1.1⁻), NKT cells (CD45⁺/CD3⁺/NK1.1⁺), and NK cells (CD45⁺/CD3⁻/NK1.1⁺) in indicated organs ($n \geq 4$ mice per group). **j** Fibrotic areas in indicated organs based on the Sirius Red-stained sections in Supplementary Figure 6b. ($n = 5$ mice per group). Student's $t$-test was used for all comparisons between $Prf1^{-/-}$ mice and WT mice. **k** Differentially expressed genes in indicated organs. Notable enriched GO terms are highlighted ($n \geq 3$ mice per group). **l** SASP genes score, presented as expression mean log ratio of old ABT-737 and Vehicle treated over young $Prf1^{-/-}$ mice (Supplementary Figure 6c). The bottom panel depicts number of detected SASP genes. ($n \geq 3$ mice per group). **m** Upper panel: Euler diagram of differentially expressed genes from all analyzed tissues. Lower panel: Log ratio of gene expression for 20-months-old ABT-737 ($y$ axis) and Vehicle ($x$ axis)-treated over young $Prf1^{-/-}$ mice. Dashed line represents linear regression fitted to the data; $p$ value computed against the null hypothesis that tilt = 1; ($n \geq 3$ mice per group). For all graphs, values are means ± SEM. Student's $t$-test was used for all other comparisons. (*$P < 0.05$, **$P < 0.01$, ***$P < 0.001$)

the activation levels of NF-κB in the liver. The number of cells that are positive for nuclear p65 was more than two-fold decreased in the ABT-737 treated mice comparing to the control mice ($P = 0.019$; Fig. 5e, f). To examine the systemic effect of ABT-737 treatment in these mice we assayed cytokines in their serum using a cytokine array (Fig. 5g, Supplementary Figure 7d). Interestingly, the amounts of the eight cytokines that had been among the most strongly upregulated in old $Prf1^{-/-}$ mice were reduced upon ABT-737 treatment in $LMNA^{+/G609G}$ mice (Fig. 5g, Fig. 2a). We next determined the lifespan of $LMNA^{+/G609G}$ mice treated with ABT-737 or with a vehicle. Strikingly, ABT-737-treatment resulted in a significant increase in median survival (377 vs. 353 days, $P = 0.0007$), compared with vehicle-treated mice (Fig. 5h). ABT-737 effect on lifespan was observed in both male and female $LMNA^{+/G609G}$ mice (Fig. 5h). The observed increase in median survival in the ABT-737-treated mice was accompanied by a delayed decrease in weight loss during their late stages of life (Fig. 5i). Our data thus suggest that the treatment with ABT-737 eliminates senescent cells, reduces the levels of circulating cytokines, and extend the median lifespan in $LMNA^{+/G609G}$ mice.

**Perforin deficiency shortens the lifespan of progeroid mice.** The involvement of immune cell cytotoxicity in the regulation of the presence of senescent cells and in accelerated aging in progeroid mice is largely unknown. By crossing $Prf1^{-/-}$ mice with $LMNA^{+/G609G}$ mice, we generated $LMNA^{+/G609G}/Prf1^{-/-}$ mice, $LMNA^{+/G609G}/Prf1^{+/-}$ mice, and $LMNA^{+/G609G}/Prf1^{+/+}$ mice, and followed their survival. In a similar manner to $Prf1^{-/-}$ mice, $LMNA^{+/G609G}/Prf1^{-/-}$ mice exhibited reduced median survival compared to control $LMNA^{+/G609G}/Prf1^{+/+}$ mice, suggesting that immune cell cytotoxicity acts to restrain accelerated aging in progeroid mice (Fig. 6a). Intrigued by this finding, and by the finding that in $LMNA^{+/G609G}$ mice ABT-737 treatment reduced the presence of cytokines upregulated in old $Prf1^{-/-}$ mice, we asked whether clearance of senescent cells could counteract accelerated aging which stems from a combination of progerin accumulation and impaired immune cell cytotoxicity. For this purpose, we administered ABT-737 or vehicle to $LMNA^{+/G609G}/Prf1^{-/-}$ female mice two consecutive days monthly, since the age of 7-month until the age of 11 months (Fig. 6b). In line with the accumulation of senescent cells in $Prf1^{-/-}$ mice, selected tissues from $LMNA^{+/G609G}/Prf1^{-/-}$ mice showed a more than 2-fold increase in the numbers of SA-β-Gal + cells when compared to $LMNA^{+/G609G}$ control mice, (Fig. 6c, d). As expected, vehicle-treated $LMNA^{+/G609G}/Prf1^{-/-}$ mice were not different from the untreated mice of the same age. Importantly,

ABT-737 treatment in $LMNA^{+/G609G}/Prf1^{-/-}$ mice caused a significant decrease in the numbers of SA-β-Gal + cells, bringing them close to the levels of these cells in the $LMNA^{+/G609G}$ control mice (Fig. 6c, d). Accumulation of senescent cells in tissues of $LMNA^{+/G609G}/Prf1^{-/-}$ mice could intensify the systemic inflammatory response that is driving the accelerated aging phenotype of $LMNA^{G609G}$ mice[50]. Indeed, staining of livers from $LMNA^{+/G609G}/Prf1^{-/-}$ mice showed an increase in the numbers of cells that were stained positively for nuclear p65 (Fig. 6e, f). Relative to $LMNA^{+/G609G}/Prf1^{+/+}$ mice, the $LMNA^{+/G609G}/Prf1^{-/-}$ mice had an increased WBC count (Fig. 6g) and enlarged spleen (Fig. 6h). Administration of ABT-737 was not only able to alleviate all of those aspects of inflammation (Fig. 6e–h), but also reconcile compromised integrity and tissue fibrosis that are associated with high inflammatory load in the tissues (Fig. 6i). Lastly, by carrying out the same ABT-737 administration protocol while monitoring survival, we found that ABT-737 treatment extended the median lifespan in both male and female $LMNA^{+/G609G}/Prf1^{-/-}$ mice (Fig. 6j). Our data thus indicate that $Prf1$ deficiency in progeroid mice further escalates their aging process by promoting the early establishment of chronic inflammation. Pharmacological clearance of senescent cells by ABT-737, on the other hand, is able to halt deleterious consequences of impaired cell cytotoxicity and increase survival in this model.

## Discussion

Senescent cells accumulate with age and contribute to aging and age-associated pathologies[36,41,51–55]. In a variety of physiological conditions in young organisms, senescent cells are subjected to immune surveillance[6,7,9,17,22,56–58]. The extent to which the dis-regulation of such immune surveillance contributes to senescent cell accumulation in the aging organism is not known. In this study we used mouse models to investigate the long-term consequences of deficiency in the expression of perforin, a key mediator of immune cell cytotoxicity, and to evaluate its implications for compromised immune functionality during aging. We showed that perforin-mediated elimination of senescent cells is essential for limiting the tissue burden of naturally occurring senescent cells throughout the aging process. The high load of senescent cells resulted from perforin deficiency was accompanied by chronic inflammation, which in turn strongly affected morbidity and mortality. Pharmacological elimination of senescent cells from these mice alleviated their age-related phenotypes. Perforin deficiency also caused an increase in the senescent-cell load that accumulates early in life owing to mutation of the $LMNA$ gene. Pharmacological elimination of senescent cells from

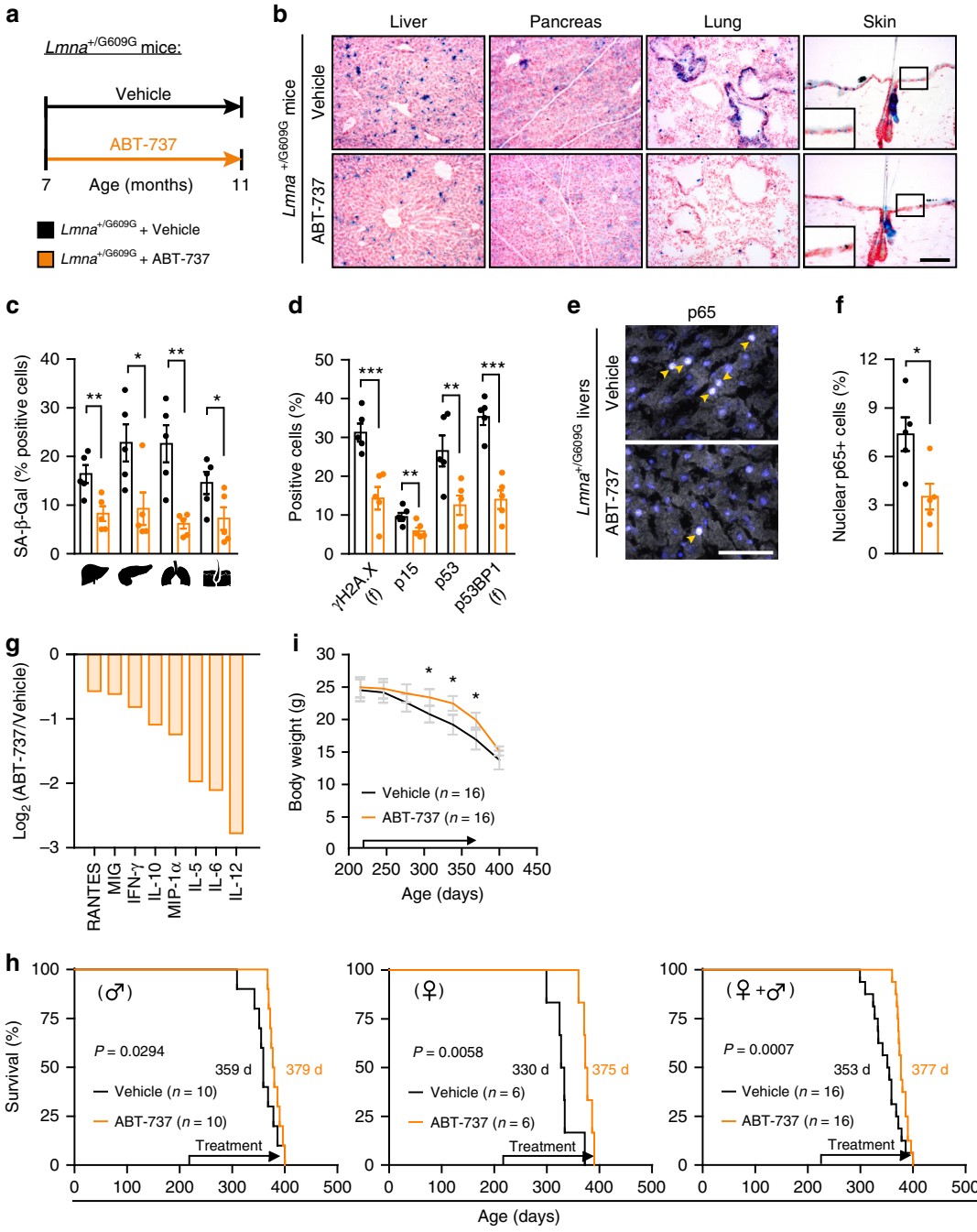

**Fig. 5** Treatment with ABT-737 increases median lifespan of progeroid mice. Starting the age of 7 months ABT-737 or DMSO-based vehicle solution was administered to *LMNA*+/G609G mice. **a** Scheme of drug administration to *LMNA*+/G609G mice. **b** Representative images depicting SA-β-Gal activity in frozen sections of livers, pancreas, lungs, and skin from ABT-737-treated and vehicle-treated *LMNA*+/G609G female mice at the age of 11 months. Scale bar, 100 μm. **c** Percentage of cells with SA-β-Gal activity in liver, pancreas, lungs, and skin from ABT-737-treated and vehicle-treated *LMNA*+/G609G female mice. (*n* = 5 mice per group). **d** Percentages of cells stained positively for γH2AX, p15, p53, p53BP1, and DcR2 in ABT-737-treated and vehicle-treated *LMNA*+/G609G female mice at the age of 11 months. (*n* = 5 mice per group). **e** Representative images of immunofluorescence staining for p65 (arrows) in liver sections from ABT-737-treated and vehicle-treated *LMNA*+/G609G female mice at the age of 11 months. Scale bar, 50 μm. **f** Percentage of p65 + nuclei in livers from ABT-737-treated and vehicle-treated *LMNA*+/G609G female mice at the age of 11 months. (*n* = 5 mice per group). **g** Serum cytokine levels in ABT-737-treated *LMNA*+/G609G female mice relative to age-matched vehicle -treated *LMNA*+/G609G female mice. Bars represent the average intensity of 3 pooled samples (log2). Values are shown for cytokines that were decreased by more than 2-fold. **h** Kaplan–Meier survival curves of ABT-737-treated and vehicle-treated *LMNA*+/G609G mice. Data is shown for males (*n* = 10), females (*n* = 6), and both genders combined (*n* = 16). Chi square Gehan–Breslow–Wilcoxon Test was used for statistical analysis. **i** Weight curves of ABT-737-treated and vehicle-treated *LMNA*+/G609G mice. (*n* = 16 mice per group). For all graphs, values are means ± SEM. Student's *t*-test was used for all comparisons between the two groups. (\**P* < 0.05, \*\**P* < 0.01, \*\*\**P* < 0.001)

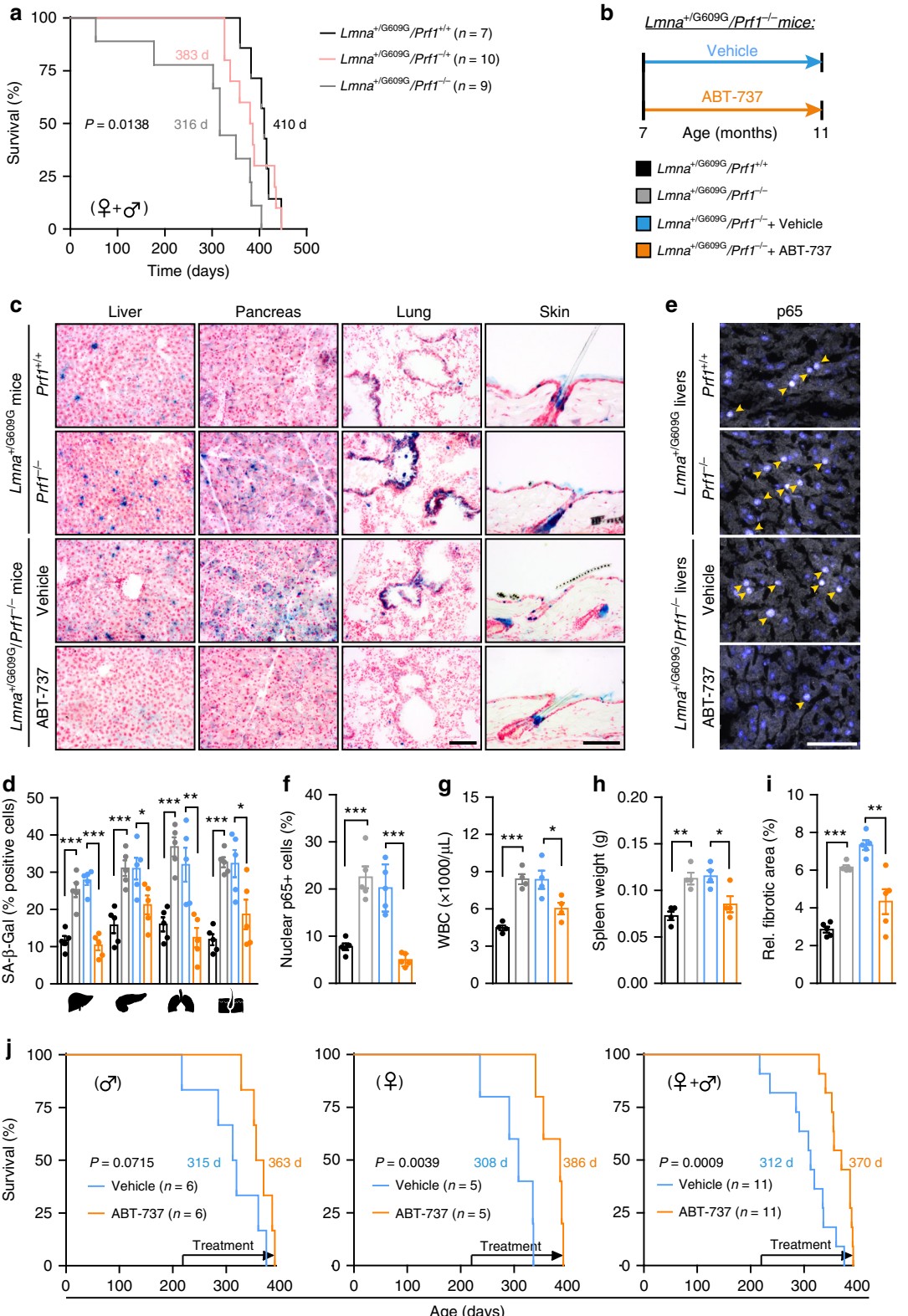

these mice extended their lifespan. These findings demonstrate the importance of immune clearance of senescent cells for the aging process, as well as the relevance and the potential promise of pharmacological elimination of senescent cells in counteracting the age-related phenotypes.

A decline in immune function is one of the natural consequences of aging[59]. Diminished cell mediated immune function

is indeed associated with the increased morbidity and mortality that result from atherosclerosis, infectious diseases, vaccine failure, and possibly also autoimmunity and cancer, whereas sustained immune function is associated with healthy aging and longevity[60]. Interestingly, reduced release and binding of perforin at the immunological synapse underlies the age-related decline in NK-cell cytotoxicity[24,60]. Therefore, our observation that

**Fig. 6** Treatment with ABT-737 increases median lifespan of progeroid mice with perforin deficiency. $LMNA^{+/G609G}/Prf1^{-/-}$ mice, $LMNA^{+/G609G}/Prf1^{+/-}$, and $LMNA^{+/G609G}/Prf1^{+/+}$ mice from both genders underwent a survival test. **a** Kaplan–Meier curves of $LMNA^{+/G609G}/Prf1^{-/-}$ mice, $LMNA^{+/G609G}/Prf1^{+/-}$ mice and $LMNA^{+/G609G}/Prf1^{+/+}$ mice. Chi square Gehan–Breslow-Wilcoxon test was used for statistical analysis. Additional cohorts of $LMNA^{+/G609G}/Prf1^{+/+}$, $LMNA^{+/G609G}/Prf1^{-/-}$, ABT-737-treated and Vehicle-treated $LMNA^{+/G609G}/Prf1^{-/-}$ female mice were raised. **b** Scheme of drug administration to $LMNA^{+/G609G}/Prf1^{-/-}$ mice. Starting the age of 7 months ABT-737 or DMSO-based vehicle solution was administered to $LMNA^{+/G609G}/Prf1^{-/-}$ female mice as described above. At the age of 11 months, mice from all groups were sacrificed and selected organs examined for the presence of senescent cells. **c** Representative images depicting SA-β-Gal activity in frozen sections of livers, pancreas, lungs, and skin from all groups of mice at the age of 11 months. Scale bar, 100 μm. **d** Percentage of SA-β-Gal + cells in livers, pancreas, lungs, and skin from all groups of mice. ($n = 4$ mice per group). **e** Representative images depicting immunofluorescence staining of p65 in livers from all mice groups. Scale bar, 100 μm. **f** Percentage of p65 + nuclei in livers from all groups. ($n = 4$ mice per group). **g** White blood cell counts of mice from all groups. ($n = 4$ mice per group). **h** Spleen weights of from all mice groups. ($n = 4$ mice per group). **i** Percentage of fibrotic area in livers paraffin-embedded sections of 11 month mice from all groups. ($n = 4$ mice per group). For all graphs, values are means ± SEM. Student's t-test was used for all comparisons between mice groups. **j** Kaplan–Meier survival curves of ABT-737-treated and vehicle-treated $LMNA^{+/G609G}/Prf1^{-/-}$ mice. Data is shown for males ($n = 6$), females ($n = 5$), and both genders combined ($n = 11$). Chi square Gehan–Breslow–Wilcoxon test was used for statistical analysis. (*$P < 0.05$, **$P < 0.01$, ***$P < 0.001$)

senescent cells extensively accumulated in old $Prf1^{-/-}$ mice links age-related immune dysfunction to the pathogenesis of age-related pathologies.

In various pathological conditions, senescent cells can interact with different effector cells from both the innate and the adaptive immune systems[6,9,17,22,56–58]. The nature of interacting effector cells in the context of aging is still unknown[19]. Both NK cells and cytotoxic T cells might execute their cytotoxic functions via the perforin pathway. NK and T cells were present in abundance at sites of the immune foci observed in our old $Prf1^{-/-}$ mice. It is therefore conceivable that both innate and adaptive immune mechanisms play a role in the surveillance of senescent cells during aging. Immune cell accumulation in tissues might further contribute to the high level of inflammation that we observed in the perforin-deficient mice.

Senescent cells can induce an immune response via the robust secretion of various cytokines, chemokines, and growth factors[2]. These factors can cause an increase in total body inflammation, both directly and via immune cell activation. The activated immune cells might not be capable of eliminating senescent cells in $Prf1^{-/-}$ mice, but they are still potent secretors of pro-inflammatory cytokines[60]. Therefore, the age-dependent accumulation of senescent cells on the background of immune incompetence might have the deleterious result of increasing the lymphocyte numbers in the examined tissues. Blood samples from old $Prf1^{-/-}$ mice indeed showed an increase in both pro-inflammatory and anti-inflammatory signals, suggesting that the immune system is failing to reach homeostasis, and that a pro-inflammatory environment has consequently become chronic. This "inflammaging"-like status, which developed in our old $Prf1^{-/-}$ mice, might be driving accelerated aging, reduced fitness, increased weight loss and kyphosis, and leading eventually to premature death.

In $Prf1^{-/-}$ mice, cytotoxic cells are incapable of executing their function properly. Thus, in addition to senescent cells, their other target cells, such as damaged cells and cancer cells, remain in tissues and provoke pathological conditions like infections and malignancies[61]. We indeed observed that $Prf1^{-/-}$ mice display higher propensity to develop lymphomas. Still, we could not exclude the possibility that accumulation of senescent cells by itself can contribute to fueling tumor development as reported for other malignancies[62,63]. In any case, such malignancies, although lifespan-limiting, cannot account for the accelerated aging phenotypes that we observed.

Increased inflammation is also observed in HGPS patients, leading to death at the age of 13 years on average[64]. A study on the $LMNA^{+/G609G}$ murine model of HGPS showed that nuclear envelope aberrations, a hallmark of this syndrome, drive activation of the NF-κB pathway[50]. This pathway is a master regulator of inflammatory signals, and its constitutive activation eventually leads

to the establishment of chronic inflammation in those mice. In our study, senescent cell accumulation in $LMNA^{+/G609G}$ mice was enhanced upon perforin deficiency. These findings suggested that immune surveillance mechanisms in $LMNA^{+/G609G}$ mice, in which the production of senescent cells is increased, act to restrict the tissue burden of senescent cells.

Elimination of senescent cells in mice was shown to reduce incidence of age-related disorders and increase median survival[36,65]. Both ABT-737 and its paralog ABT-263 induce clearance of senescent cells in variety of in vivo systems[42,44,55]. This study shows for the first time that regardless of the functionality of the immune system, and with no genetic manipulation, it is possible to eliminate senescent cells in vivo and to modulate aging in mice by treating them with BCL-2 family inhibitors. Unfortunate possible side effects of systemic ABT-737 administration are transient thrombocytopenia and leucopenia[66,67], and we cannot exclude the possibility that the beneficial effect of ABT-737 in our study was mediated, at least in part, through its impact on the hematopoietic system.

Accumulation of senescent cells promotes aging and the development of age-related diseases; however, senescent cells also perform essential biological functions in wound healing and the response to short-term tissue damage[7,68]. To avoid interference with beneficial functions of the senescence program, interventions that target senescent cells in the context of aging and age-related diseases might need to be temporally and locally specific for particular conditions. This could be achieved through further study of specific mechanisms for immune surveillance of senescent cells. Such research will eventually lead to the development of novel therapeutic strategies for age-related pathologies. Overall, this study links the age-related decline in immune system capability to the accumulation of senescent cells during aging, and further establishes the contribution of these factors to age-related pathologies and reductions in health-span and lifespan.

## Methods
**Mice.** All mice were bred and maintained under specific pathogen-free conditions at the Weizmann Institute of Science in accordance with national animal care guidelines. All animal experiments were approved by the Weizmann Institute's Institutional Animal Care and Use Committee. Mice were maintained under a specific pathogen-free environment throughout the study. $Prf1^{-/-}$ mice were obtained from Jackson Laboratories (Bar Harbor). C57BL/6 mice (Harlan) served as WT controls. Several cohorts of 5 or more mice of both genotypes for survival and behavioral tests were established over time and observed for the length of the experiment. The new cohorts entered the experiment till reaching the designated size of the overall cohort which was designed based on the previous studies. The mice were assigned to the experimental groups according to their genotype without randomization or blinding. The mice that become obese during the first year of the experiments were excluded from further study. Equal number of mice from both genders was included in the study. After in vivo phenotyping, mice were sacrificed at the age of 2 months ("young"), 12 months ("mid age"), or 24 months ("old"). Heparinized blood and plasma were collected. The blood count and biochemistry laboratory was blind to the sample identity. Single-cell suspensions from perfused

liver, pancreas, lungs, and skin were taken for analysis by flow cytometry and ImageStream®X analysis. Alternatively, tissues were fixed in formalin and embedded in paraffin blocks, or immediately embedded in OCT and kept at −80 °C. $Lmna^{G609G}$ mice were kindly provided by Carlos López-Otin lab[50], and were crossed with $Prf1^{−/−}$ mice. All $Lmna^{+/G609G}$ mice were on a C57BL/6 genetic background.

**Elimination of senescent cells in vivo.** For selective elimination of senescent cells, 18-month-old $Prf1^{−/−}$ mice were subjected to ABT-737 treatment for 2 months. At the beginning of each month, 2 consecutive daily intra-peritoneal (i.p.) injection of ABT-737 (25 mg/kg body weight; Selleck Chemicals) or Vehicle as a control were given as described. ABT-737 and DMSO-based Vehicle were prepared in a working solution (30% propylene glycol, 5% Tween 80, 3.3% dextrose in water pH 4–5). $Lmna^{+/G609G}$ mice and $Lmna^{+/G609G}$; $Prf1^{−/−}$ mice were injected with ABT-737 from the age of 7 months. Mice injected with an equivalent concentration of DMSO served as controls.

**In vivo phenotyping.** Animals weights were monitored on monthly basis. Body composition was determined by EchoMRI (Echo Medical Systems). Voluntary exercise was recorded with a recording running wheel placed in individual cages for a week. Grip strength was assessed by a hang-wire test, which measured the time taken by the mouse to drop down from the wire. Hair density was measured by counting hair follicles on a single plane of tissue section. Gray hair was assessed under the microscope by counting the number of gray hairs per square centimeter of low back skin. Kyphosis was evaluated by scanning of each mouse with a cone-beam volumetric scanner (TomoScope ® 30 s Duo). The kyphotic index (D2/D1) was obtained from micro-CT images, where D1 is the linear distance measured from the 7th cervical to the 6th lumbar vertebra and D2 is the vertical distance from the highest point along that line to the vertebral body, corresponding to the apex of spinal curvature. The higher the ratio, the greater is the kyphosis. Blood was extracted directly from the mandibular sinus. Relative levels of plasma cytokines and chemokines were assessed with an ARY006 mouse cytokine array kit (R&D Systems).

**Histological analysis.** For SA-β-gal activity assay, frozen tissue sections were fixed with 0.5% glutaraldehyde in PBS for 15 min, washed with PBS supplemented with 1 mM MgCl$_2$, and stained for 6–8 h in X-Gal staining solution. Sections were counterstained with Nuclear Fast Red (Sigma).

Paraffin-embedded tissue sections were stained with hematoxylin and eosin (H&E) for routine examination or with Sirius Red for visualization of fibrotic deposition. For quantification of fibrosis, at least three whole sections from each mouse were scanned by Laser Scanner Cytometry (CompuCyte). These images were quantified using NIH ImageJ software (http://rsb.info.nih.gov/ij/). We calculated the amount of fibrotic tissue in old mice on the basis of the positively stained area per field.

The following antibodies were used: γ-H2AX (1:200, Cell Signaling, S139), p15 (1:200, Assay Biotech, C0287), p16 (1:400, Abcam, ab54210), p53 (1:50, Santa Cruz, sc-6243), p53BP1 (1:200, Novus, NB100-304), p65 (1:100, Cell Signaling, c-3033 s), and DcR2 (1:200, Enzo, ADI-AAP-371-E), Alexa647 anti-CD45 (1:100, #103124, Biolegend), APC anti-Gr-1 (1:100, #108412, Biolegend), APC anti-B220 (1:100, #103212, Biolegend), Alexa647 anti-NK1.1 (1:100, #108720, Biolegend), Alexa647 anti-CD3 (1:100, #100209, Biolegend). For immunostaining, tissue sections (10 μm) were mounted on slides and air-dried for 15 min, fixed in 4% paraformaldehyde (PFA; Sigma) for 10 min, and permeabilized in 0.2% Triton X-100 for 15 min. Slides were blocked in a solution of 5% goat serum, 1% BSA and 0.2% Triton X-100 for 1 h, then incubated with primary antibody overnight at 4 °C in the blocking solution. The following day the slides were washed twice with 0.2% Triton X-100 and once with PBS, each time for at least 30 min. In case primary antibodies were not fluorescently labeled, samples were incubated with Dylight-549-conjugated secondary antibody (1:300, Jackson ImmunoResearch) in blocking solution for 1 h at room temperature. After additional washings, nuclei were counterstained with DAPI (Roche) and the slides were mounted in VectaShield mounting medium (Vector Laboratories). Samples were imaged by Olympus CX41 microscope or Zeiss LSM710 confocal microscope.

**Flow cytometry and ImageStreamX analysis.** Tissues were chopped and dissociated to single-cell suspensions by incubation of samples for 50 min at 37 °C. The dissociation solution contained HBSS X1 (#14025050, Gibco) supplemented with 0.1 mg/ml DNase I (#10104159001, Roche), and 2 mg/ml dispase II (#04942078001, Roche) or 0.5 mg/ml collagenase P (C5138, Sigma) or 1 mg/ml collagenase IV (C9263, Sigma) for dissociation of liver, pancreas or lung, respectively. Cells were collected by filtration through with 100-μM filter mesh, and washed twice with PBS.

For characterization of immune subsets in the tissues by flow cytometry, the cells were maintained in cold FACS buffer (PBS containing 1% FCS and 0.02% Sodium Azid) throughout all procedure. Cells were incubated with FITC anti-CD45 (#103108, Biolegend), PE anti-NK1.1 (#108708, Biolegend), and Alexa647 anti-CD3 (#100209, Biolegend) mixture for 30 min in 4 °C. The dilution of these

antibodies was 1:100 for the studies. After wash, DAPI was shortly introduced in order to exclude dead cells. Cells were analyzed in a SORP-LSRII instrument (BD Biosciences) and FlowJo v10 software. The cell gating strategy is shown in the Supplementary Figure 8a.

For analysis by ImagestreamX cells were first fixed in 4% PFA for 5 min. Fixed cells were washed once with PBS and then with PBS/1 mM MgCl$_2$ at pH 5.5. They were then resuspended in 5 ml of freshly prepared X-Gal staining solution,and incubated at 37 °C for 12 h while sealed and protected from light. After the staining, the cells were washed and fixed with fixation buffer for 30 min at 4 °C, washed twice with permeabilization buffer (both eBioscience), and incubated overnight with the primary antibody anti-HMGB1 (1:200, Ab18256, Abcam), washed twice and incubated for 45 min with secondary antibodies (Jackson ImmunoResearch). For identification of immune cells, the samples were incubated for 1 h with Brilliant Violet (BV605)-labeled CD45 (1:100,#103140, Biolegend). The cells were then washed, stained for 10 min with DAPI, and imaged by ImageStreamX flow cytometry (Amnis).

At least $3 × 10^4$ cells were collected from each sample. Images were analyzed using IDEAS 6.1 software (Amnis). The cell gating strategy is shown in the Supplementary Figure 8c. Gating for single cells was done using the area and aspect ratio features on the bright field (BF) image, and for focused cells using the contrast and gradient RMS features. SA-β-Gal staining of cells was quantified in the BF channel. For nuclear HMGB1 or CD45 quantification, the spot-count feature was applied on a spot mask created for the relevant acquisition channel, separating the bright spots from the background.

**Quantitative PCR.** Livers were harvested following perfusion and were flash frozen in −80 °C until further processing. Total mRNA from liver was purified using RNeasy kit (74104, Qiagen) according to manufacturer's guidelines. cDNA was reverse transcribed using random hexamers, and amplified using Platinum SYBR Green qPCR SuperMix (11744-500, Life Technologies) in a StepOnePlus™ Real-Time PCR System (Applied Biosystems). Relative expression was normalized using the expression levels of $GAPDH$. Primer sequences are available in the spplementary methods.

**Preparation of libraries for RNA-seq.** Tissues were harvested following perfusion and submersed whole in RNAlater Solutions (Invitrogen). Samples were flash frozen in −80 °C until further processing. mRNA from all tissues was purified using RNeasy mini kit (Qiagen) according to manufacturer's guidelines. RNA quality was assessed by TapeStation (Agilent). Libraries were prepared using a derivation of MARS-seq[69], optimized for bulk RNA-seq. Briefly, 50 ng of RNA from each sample were reverse transcribed into cDNA, pooled and linearly amplified by in vitro transcription. Following fragmentation and addition of Illumina sequences, successful final libraries were sequenced using the Illumina NextSeq-500 platform.

**Statistical analysis.** Results are presented as means ± SDs. Differences between mouse cohorts were analyzed with a two-tailed Student's $t$-test; values of $P < 0.05$ were considered significant. Quantitative differences between positive nuclei in indirect immunofluorescence experiments were analyzed by ANOVA. Results of the grip strength test were analyzed by ANCOVA, with body mass as the covariate. For the statistical analysis of survival Chi square Gehan–Breslow–Wilcoxon Test was used. The varience was similar between the experimetal groups. The analysis was performed using GraphPad Prism software.

## Data availability

The authors declare that the data supporting the findings of this study are available within the paper and its Supplementary Information files. A Reporting Summary for this Article is available as a Supplementary Information file. RNA sequencing data analyzed in this publication have been deposited in NCBI's Gene Expression Omnibus[70], and are accessible through GEO Series accession number GSE115401. Other data that support the findings of this study are available from the corresponding author upon reasonable request.

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

## Acknowledgements

We thank Carlos López-Otin (Universidad de Oviedo, Spain) for the Lmna^G609G mice. We thank all members of the Krizhanovsky Laboratory for critical reading of the manuscript and insightful discussions. We thank Ori Brenner (WIS) for his assistance in pathological evaluation of old mice, and Inbal Biton (WIS) for assisting in micro-CT analysis. H.G and R.S were supported by RTGCEMMA funded by the DFG. Y.O. was supported by German-Israeli

Helmholtz Research School. This work was supported by grants to V.K. from the European Research Council under the European Union's FP7, and from the Israel Science Foundation.

## Author contributions

Y.O., T.L., and H.L. carried out experiments and analyses. E.V. was in charge of animal breeding. Y.O. and E.V. performed surgical procedures on mice. H.Gal., J.W., and A.Shapira performed experiments. A.B., R.Y., A.Sagiv., R.S., and A.A participated in designing experiments. M.T. participated in design and execution of mouse fitness studies. H.G., I.A. supervised part of experiments and analysis. Y.O. and V.K. designed the study and wrote the manuscript. V.K. supervised the project.

## Additional information

**Competing interests:** The authors declare no competing interests.

