## [Peer Review File · Nature Communications]

Reviewer #1 (Remarks to the Author):

This is a revised version of a manuscript that was originally reviewed for Nature Medicine. The authors have revised their manuscript, added new data and the data is much more compelling. In my view, the manuscript should be published in Nature Communications.

Reviewer #2 (Remarks to the Author):

The authors present a heavily reworked manuscript, including significant amounts of new data. In particular, they adequately address the principle concerns, including the demonstration of time-course changes in senescent-cell accumulation in WT and Prf^{-/-} mice, further characterisation of the immune-capacity of Prf^{-/-} mice, and reworking the text to clarify many points.

One additional comment that should be included in light of new data is to mention in which cell types the p16 and also the SABgal staining, is seen in the liver (figs 1d and 1f) as it is not clear that the staining is actually in hepatocytes or other cells.

This reviewer congratulates the authors on a significantly improved manuscript.

Reviewer #3 (Remarks to the Author):

The revised version of this manuscript alleviated the major concerns raised from the initial submission. The tempering of conclusions to reflect that shown in this study and the inclusion of additional information related to the models is appreciated. The revised version represents a substantial improvement.

Ovadya et al.

Point-by-point response letter – second revision.

We thank all the reviewers for their positive view on our revised manuscript.

Reviewer #1 (Remarks to the Author):

This is a revised version of a manuscript that was originally reviewed for Nature Medicine. The authors have revised their manuscript, added new data and the data is much more compelling. In my view, the manuscript should be published in Nature Communications.

We are glad to see that there is no further comments from this reviewer

Reviewer #2 (Remarks to the Author):

The authors present a heavily reworked manuscript, including significant amounts of new data. In particular, they adequately address the principle concerns, including the demonstration of time-course changes in senescent-cell accumulation in WT and Prf^{-/-} mice, further characterisation of the immune-capacity of Prf^{-/-} mice, and reworking the text to clarify many points.

One additional comment that should be included in light of new data is to mention in which cell types the p16 and also the SABgal staining, is seen in the liver (figs 1d and 1f) as it is not clear that the staining is actually in hepatocytes or other cells.

This reviewer congratulates the authors on a significantly improved manuscript.

We are happy to see that the reviewer notices that our manuscript is significantly improved. The reviewer is interested to reveal in what cells the SA-b-gal staining in the liver occurs. The revised text and the existing data of the manuscript address this point. The text of the Results section now states that the cells are mainly non-hepatocytes (ms, p5 second paragraph). The experiments described later in the manuscript, using single cell approach (Fig. 1i) show that these cells are also part of CD45⁻ population, further supporting that they are resident non-immune cells. The studies of exact populations that become senescent in the liver during aging were recently published and as such the detailed examination of these cell populations is beyond the scope of our study.

Reviewer #3 (Remarks to the Author):

The revised version of this manuscript alleviated the major concerns raised from the initial submission. The tempering of conclusions to reflect that shown in this study and the inclusion of additional information related to the models is appreciated. The revised version represents a substantial improvement.

We are glad to see that there is no further comments from this reviewer